# Low-Molecular-Weight β-1,3-1,6-Glucan Derived from *Aureobasidium pullulans* Exhibits Anticancer Activity by Inducing Apoptosis in Colorectal Cancer Cells

**DOI:** 10.3390/biomedicines11020529

**Published:** 2023-02-11

**Authors:** Ji Hyeon Kim, Jeonghyeon Seo, Huiwon No, Takao Kuge, Takahiro Mori, Hisashi Kimoto, Jin-Kyung Kim

**Affiliations:** 1Department of Biomedical Science, Daegu Catholic University, Gyeongsan-si 38430, Republic of Korea; 2ADEKA Corporation, R&D Division, 7-2-34, Higashi-Ogu, Arakawa-ku, Tokyo 116-8553, Japan; 3Graduate School of Bioscience and Biotechnology, Fukui Prefectural University, 88-1 Futaomote, Awara-shi 910-4103, Japan

**Keywords:** *Aureobasidium pullulans*, β-1,3-1,6-glucan, colorectal cancer, apoptosis

## Abstract

β-glucan, a plant polysaccharide, mainly exists in plant cell walls of oats, barley, and wheat. It is attracting attention due to its high potential for use as functional foods and pharmaceuticals. We have previously reported that low-molecular-weight *Aureobasidium pullulans*-fermented β-D-glucan (LMW-AP-FBG) could inhibit inflammatory responses by inhibiting mitogen-activated protein kinases and nuclear factor-κB signaling pathways. Bases on previous results, the objective of the present study was to investigate the therapeutic potential of LMW-AP-FBG in BALB/c mice intracutaneously transplanted with CT-26 colon cancer cells onto their backs. Daily intraperitoneal injections of LMW-AP-FBG (5 mg/kg) for two weeks significantly suppressed tumor growth in mice bearing CT-26 tumors by reducing tumor proliferation and inducing apoptosis as compared to phosphate buffer-treated control mice. In addition, LMW-AP-FBG treatment reduced the viability of CT-26 cells in a dose-dependent manner by inducing apoptosis with loss of mitochondrial transmembrane potential and increased activated caspases. Taken together, LMW-AP-FBG exhibits anticancer properties both in vivo and in vitro.

## 1. Introduction

*Aureobasidium pullulans* is well known as a black yeast. It can extracellularly produce β-1,3-1,6-glucan under certain conditions [1]. Several *A. pullulans* strains including ADK-34 can extracellularly secrete large quantities of a branched type of β-1,3-1,6 glucan, so called *A. pullulans*-fermented β-D-glucan (AP-FBG) [1,2]. To date, various physiological activities and therapeutic effects of AP-FBG have been reported. For example, intraperitoneal injection of AP-FBG can accumulate leucocytes and neutrophils and promote production of tumor necrosis factor (TNF)-α, interleukin (IL)-12, and interferon-γ from peritoneal exudate cells [3,4]. AP-FBG also exhibits antitumor and immunostimulatory activities. It has already been used as a biological response modifier as well as a functional food [1,5]. As described above, AF-FBG has high utilization value in various fields, but has a problem in that it is difficult to use efficiently due to high viscosity and difficult drying and concentration [1]. To overcome these problems, a method for reducing the viscosity and molecular weight of AP-FBG using mechanochemical ball milling was developed to produce low molecular weight AP-FBG (LMW-AP-FBG) [6]. We also demonstrated that LMW-AP-FBG exerts anti-inflammatory effects in murine RAW264.7 macrophages through inhibition of mitogen-activated protein kinases (MAPK) and nuclear factor-κB (NF-κB) signaling pathways [6].

In the last few decades, the anticancer effects of polysaccharides, including β-glucan, have been elucidated [1,7,8]. It is also well known that the anticancer response of polysaccharides is dependent on their origin, structure and composition [1,7,8]. For example, laminarin, a water-soluble polysaccharide present in brown algae, exhibits anticancer activity in retinoblastoma Y79 cells by inducing apoptosis and arresting the cell cycle in the G2/M phase [9]. The oat β-glucan exerted cytotoxic action on human skin melanoma, HTB-140 cells [10]. These oat and barley β-glucans are predominantly linear, with large regions of 1,4-β-glucan molecules separating short stretches of 1,3-β-glucan molecules. In addition, Kimura et al. showed the antitumor and antimetastatic activities of AP-FBG using a mouse model in which CT26 cells were transplanted into the spleen [5]. 

Colorectal cancer (CRC) is one of the most common cancers in the world. The number of CRC patients worldwide was 1.8 million in 2018, the third highest number of patients with malignant neoplasms [11]. Inflammation is a physiological response to healing of damaged tissue. It is closely related to tumorigenesis [12,13]. Inflammatory responses begin with excretion of various biomolecules from damaged tissues. Subsequently, immune cells trek to injured tissues to fix the injury. When the wound has healed, the inflammatory signaling cascade has completed. Besides, chronic inflammation results from activation of signaling pathways without being encouraged by injury. It is well known that patients with chronic inflammatory bowel diseases such as ulcerative colitis and Crohn’s disease have an increased risk of CRC [14]. Many previous studies have suggested that there is a strong correlation between inflammation and cancer [12,13,14]. Since our previous study showed that LMW-AP-FBG suppressed inflammatory responses, this study was conducted to investigate anticancer effects of LMW-AP-FBG in CT-26 cells, a colon carcinoma cell line generated from BALB/c mice, and CT-26 tumor-bearing mice. To the best of our knowledge, this is the first study to examine anticancer effects of LMW-AP-FBG.

## 2. Materials and Methods

### 2.1. Materials

All reagents including 5-fluorouracil (5-FU) were purchased from Sigma-Aldrich (St. Louis, MO, USA) unless otherwise noted. LMW-AP-FBG was prepared as previously described [6]. Antibodies (Abs) used in this study included Cytochrome C (sc-7159), Bcl-xL (sc-7195), proliferating cell nuclear antigen (PCNA, sc-56) obtained from Santa Cruz Biotechnology (Santa Cruz, CA, USA), BAX (#2772) from Cell Signaling Technology (Danvers, MA, USA), and GAPDH (MA5-15738) from Invitrogen (Waltham, MA, USA). 

### 2.2. Cells and Cell Viability Assay

CT-26 cells (a mouse colon cancer cell line derived from colon tissue of BALB/c mouse) were obtained from the Korea Cell Bank (Seoul, Republic of Korea). They were cultured in Dulbecco’s modified Eagle’s medium (DMEM, Hyclone, Logan, UT, USA) supplemented with heat-inactivated fetal bovine serum (FBS, Hyclone) and penicillin–streptomycin mixed solution (Hyclone) at 37 °C with 5% CO_2_. Cells were seeded into 96-well plates at a density of 5 × 10^3^ cells/well and incubated at 37 °C with 5% CO_2_ for 24 h. Cells were treated with LMW-AP-FBG (diluted in complete medium) for 24 h. Thereafter, 10 μL CCK-8 (Dojindo, Tokyo, Japan) was added followed by incubation for another 2 h. The absorbance at 450 nm was then measured with a microplate reader (Tecan, San Jose, CA, USA).

### 2.3. Measurement of Apoptotic Cells

CT-26 cell apoptosis was assayed using a Muse™ Annexin V and Dead Cell Kit (Luminex, Austin, TX, USA) according to the user’s guide. CT-26 cells grown in 12-well plates (5 × 10^4^ cells) were treated with LMW-AP-FBG at different concentrations (0, 2.5, 5, and 10 mg/mL) for 24 h. Adherent cells were washed, collected, and incubated with Annexin V and 7-AAD (7-Aminoactinomycin D-dead cell marker) for 20 min in the dark at room temperature. Percentages of live, dead, early, and late apoptotic cells were measured with a Muse^®^ Cell Analyzer (Luminex).

### 2.4. Multi-Caspase Assay

CT-26 cells cultivated with various concentration of LMW-AP-FBG for 24 h were collected to analyze the activation of multi-caspase (caspase-1, -3, -4, -5, -6, -7, -8 and -9). Cells were suspended in 1 × caspase buffer and 50 μL of Muse™ Multi-Caspase reagent working solution (Luminex). After incubating at 37 °C for 30 min, Muse™ Caspase 7-AAD (125 μL) was further added to each sample. Cells were then measured with a Muse™ Cell Analyzer (Luminex).

### 2.5. Determination of Mitochondrial Membrane Potential 

CT-26 cells (1 × 10^5^ cells) were seeded into 24-well cell culture plates and treated with LMW-AP-FBG at concentrations of 0, 2.5, 5, and 10 mg/mL for 6 h. Following harvest, cells were centrifuged at 4000 rpm, 4 °C for 5 min and washed with 1 × assay buffer. In a dark condition, cells were added with Mitopotential working solution (Luminex) and incubated at 37 °C for 20 min. After adding 7-AAD, cells were cultivated at room temperature for 10 min. Mitochondrial membrane potential was then assayed using a Muse™ cell analyzer (Luminex).

### 2.6. Western Blot Analysis

CT-26 cells (2 × 10^6^ cells) grown in culture medium on 90 mm plates were treated with indicated concentrations of LMW-AP-FBG for 24 h. Cells were rinsed with ice-cold PBS, followed by addition of 100 μL of PRO-PREP™ protein extraction solution (Seongnam, Republic of Korea) containing a fresh mixture of complete protease and phosphatase inhibitors (Roche, Switzerland) to each sample. After 30 μg of proteins were separated by 10 or 12% sodium dodecyl sulfate-polyacrylamide gel electrophoresis, they were transferred to a PVDF membrane (ATTO, Tokyo, Japan). Target protein was immunoblotted with the indicated primary antibody at 4°C overnight and then incubated with an HRP-conjugated secondary antibody at 4°C for 3 h. Protein bands were then visualized using an enhanced chemiluminescent reagent (Thermo Fisher Scientific, Waltham, MA) and DAVINCH Chemi CAS-400SM (Davinch-k, Seoul, Republic of Korea). Protein levels were analyzed using ImageJ software ver. 1.150i.

### 2.7. Animal Study

Experimental protocol for the syngeneic tumor model was carefully reviewed based on the ethical and scientific care guidelines. It was approved by Daegu Catholic University-Institutional Animal Care and Use Committee (Approval No. CUD-2021-035).

To evaluate tumorigenicity after treatment with LMW-AP-FBG, CT-26 cells (1 × 10^6^) were resuspended in 100 µL of PBS and subcutaneously injected into six-week-old female BALB/c mice (Hyochang Science, Daegu, Republic of Korea). When the tumor attained a size of 100 mm^3^, mice were divided into three groups (6 mice per group) and randomized to different treatments by intraperitoneal injection as follows: phosphate buffer (PBS), 5-FU (50 mg/kg, two times per week), and LMW-AP-FBG (5 mg/kg, every day). Tumor sizes and body weights of mice were monitored every two days by an investigator who was blind to the treatment condition. The tumor volume was measured with Vernier calipers (Mitutoyo, Japan) and calculated using the following formula: tumor = ab^2^/^2^ in mm^3^, where a and b were the longest and the shortest perpendicular diameters of the tumor, respectively. After 14 days of treatment, animals were sacrificed and tumors were taken for further analysis. Histopathology, terminal deoxyribonucleotidyl transferase-mediated dUTP-digoxigenin nick end-labeling (TUNEL, R&D Systems, Minneapolis, MN, USA), and PCNA staining were performed as previously described [15]. 

### 2.8. Statistical Analysis

Values are expressed as mean ± SEM of results. Comparisons were performed by one-way analysis of variance (ANOVA) or paired and unpaired *t* test when appropriate. Bonferroni’s correction for multiple comparisons was used to determine the level of significance using GraphPad Prism 6.0 software (GraphPad Software Inc., San Diego, CA, USA). Statistical significance was considered at *p* < 0.05.

## 3. Results

### 3.1. LMW-AP-FBG Reduces Growth of Syngeneic Transplanted CT-26 Tumors in Mice

Since our previous study has shown that LMW-AP-FBG has anti-inflammatory activity in LPS-stimulated mouse RAW264.7 macrophages, we examined the anticancer effect of LMW-AP-FBG in the present study. To investigate whether LMW-AP-FBG treatment could inhibit tumor progression in vivo, a syngeneic transplanted mouse tumor model was used. There were no significant changes in body weights following treatment with LMW-AP-FBG compared to the PBS-treated group, indicating that LMW-AP-FBG treatment did not have a toxic effect. Unlike LMW-AP-FBG treatment group, the 5-FU treatment group showed a significant weight loss from the 6th day after treatment (Figure 1a), suggesting that 5-FU treatment can induce toxicity in mice, similar to results of other studies [16]. Tumors grew progressively in PBS-treated control group as shown in Figure 1b. Intraperitoneal administration of LMW-AP-FBG (5 mg/kg body weight) significantly inhibited CT-26 colon cancer growth as evidenced by tumor volume (Figure 1b) and tumor weight (Figure 1c,d). The average tumor weight of the PBS-treated group was 1538 ± 172.0 mg/mice, while that of LMW-AP-FBG treated group was 532.1 ± 108.8 mg/mice (Figure 1c) at day 14 after treatment, indicating that LMW-AP-FBG significantly inhibited tumor growth. In addition, 5-FU treated mice showed significantly reduced tumor sizes and tumor volume, consistent with other investigations [17]. 

We next tested whether LMW-AP-FBG treatment affected cell proliferation and apoptosis in tumor tissues. LMW-AP-FBG treatment significantly reduced PCNA-positive cells (markers of proliferating cells) in tumor tissues as shown in Figure 1f,h compared to PBS treatment as a control group. In addition, LMW-AP-FBG intervention affected apoptosis as measured by TUNEL assay. A large number of TUNEL-positive cells appeared in tumor tissues of LMW-AP-FBG and 5-FU treatment group, whereas the number of TUNEL-positive was significantly lower in PBS-treated group (Figure 1g,i). These data suggest that LMW-AP-FBG treatment could block cell proliferation and promote apoptosis in vivo.

### 3.2. LMW-AP-FBG Reduces Viability of CT-26 Colon Cancer Cells

Because LMW-AP-FBG treatment reduced tumor sizes in mice bearing CT-26 colon cancer cells, we next investigated whether LMW-AP-FBG affected viability of CT-26 cells. CT-26 cells were treated with different concentrations (0, 1.25, 2.5, 5 and 10 mg/mL) of LMW-AP-FBG for 24 h. As shown in Figure 2, LMW-AP-FBG treatment reduced the viability of CT-26 cells in a dose-dependent manner. In contrast, our previous study showed that LMW-AP-FBG treatment did not affect the viability of RAW264.7 cells at a concentration up to 10 mg/mL [6]. These results suggest that LMW-AP-FBG might be more cytotoxic to CT-26 colon cancer cells than to macrophages.

### 3.3. LMW-AP-FBG Induces Apoptosis in CT-26 Cells 

To determine if apoptosis mediated the reduction of cell viability, loss of phosphatidylserine asymmetry, a hallmark of apoptosis, was quantified by cytometry using Annexin V-stained cells. Muse™ Annexin V detection assay revealed a significant increase in early apoptosis after treatment with 10 mg/mL (26.6 ± 1.14%) of LMW-AP-FBG in comparison with treatment with the medium control (14.2 ± 0.87%) as shown in Figure 3a,b. Additionally, late apoptotic cells were dramatically increased by treatment with LMW-AP-FBG at 5 mg/mL (13.7 ± 0.81%) and 10 mg/mL (12.9 ± 1.24%) in comparison with treatment with the medium as control (7.5 ± 1.24%, Figure 3a,c). These results indicate that LMW-AP-FBG treatment can reduce cell viability via inducing apoptosis.

Next, to examine whether caspase activity was related to apoptosis in LMW-AP-FBG-treated CT-26 cells, a multi-caspase assay was carried using a Muse™ Cell Analyzer. Activation of multi-caspase was significantly increased by treatment with LMW-AP-FBG at 10 mg/mL, whereas treatment with LMW-AP-FBG at 2.5 or 5 mg/mL tended to increase activation of multi-caspase without showing statistical significances (Figure 3d,e). These results indicate that activation of multi-caspases by LMW-AP-FBG could provoke apoptosis in CT-26 cells.

### 3.4. LMW-AP-FBG Reduces Mitochondrial Membrane Potential in CT-26 Cells 

Apoptosis is often accompanied by a decrease in mitochondrial membrane permeability. Therefore, mitochondrial membrane potential measurement was performed to determine whether mitochondrial-mediated signaling pathway was involved in apoptosis induced by LMW-AP-FBG using a Muse MitoPotential Kit. After treatment for 6 h, the mitochondrial membrane potential was reduced by LMW-AP-FBG in a dose-dependent manner as shown in Figure 4a,b. In CT-26 cells, the percentage of depolarized cells was 39.2 ± 1.9% in the group treated with 10 mg/mL of LMW-AP-FBG in comparison to the medium control (28.80 ± 2.9%) as shown in Figure 4a,b. These results suggest that LMW-AP-FBG-induced apoptosis is mitochondria-dependent manner.

Finally, expression levels of proteins related to mitochondria membrane potential were analyzed by Western blot analysis (Figure 4c–e). The expression level of anti-apoptotic protein Bcl-xL was downregulated by treatment with LMW-AP-FBG in CT-26 cells (Figure 4c). In contrast, the expression level of a pro-apoptotic protein Bax was induced by treatment with LMW-AP-FBG in a concentration-dependent manner (Figure 4d). Consistent with these results, levels of cytochrome c were enhanced in response to an increase in LMW-AP-FBG concentration (Figure 4e). These results suggest that one of the mechanisms of action involved in the apoptosis induced by LMW-AP-FBG is mitochondrial-mediated apoptosis.

## 4. Discussion

β-glucan has long received considerable interest due to its potent antitumor and immunomodulatory activities. Nevertheless, purification difficulties, structural heterogeneity, and low solubility have hindered the development of structure-physical relationships and their transformation into therapeutic applications. To overcome these problems, various enzymatic, chemical, and mechanical methods have been adopted to improve the purification and solubility of β-glucan [1,6,18]. As part of that effort, our group has developed methods to reduce viscosity and molecular weight of AP-FBG using mechanochemical ball-milling, produced LMW-AP-FBG, and proved its anti-inflammatory effects in murine macrophages [6]. Based on our results that LMW-AP-FBG has a strong anti-inflammatory action in mouse macrophages, we hypothesized that LMW-AP-FBG could exhibit anticancer activity. We verified that hypothesis using CT-26 cells and CT-26-bearing mice in the present study. 

LMW-AP-FBG treatment obviously attenuated tumor growth both in vivo (mice bearing CT-26 cells) and in vitro (CT-26 cells). In addition, TUNEL-positive cells and Annexin-V-positive cells were significantly increased in CT-26-bearing mice and CT-26 cells, respectively, indicating that apoptosis was significantly induced by LMW-AP-FBG treatment. These observations show that the anticancer effects of LMW-AP-FBG in colon cancer are partially facilitated by the induction of apoptosis. 

Kawata et al. [19] have demonstrated that stimulation with β-glucan produced by *A. pullulans* (AP-PG) can induce the expression of TNF-related apoptosis inducing ligand (TRAIL) in mouse and human macrophage-like cell lines. TRAIL is known to be a cytokine that can specifically induce apoptosis in transformed cells, but not in untransformed cells [20]. Kawata et al. also observed that caspases were activated in HeLa cells after treatment with supernatant of cultured medium from AP-PG-stimulated Mono Mac 6 cells, but were inhibited by an anti-TRAIL neutralizing antibody [19]. These results suggest that stimulation by AP-PG can effectively induce TRAIL in macrophages and may be related to the induction of apoptosis in tumor cells. Similarly, LMW-AP-FBG treatment induced caspase activation and impaired mitochondrial membrane potential.

Bcl-xL and Bcl-2 are anti-apoptotic proteins that can suppress apoptosis by preventing the formation of mitochondrial pores, protecting membrane integrity, and inhibiting release of cytochrome c [21,22]. In contrast, Bax as one of the pro-apoptotic proteins can enter the mitochondria and trigger apoptosis, leading to the release of cytochrome c [21,22,23]. In addition, the ratio of anti-apoptotic proteins/pro-apoptotic proteins is an important indicator that can induce the apoptosis progress of cancer cells. Cytochrome c can activate caspase protein and stimulate apoptosis [21,22,23,24]. In the present work, results showed that LMW-AP-FBG significantly down-regulated Bcl-xL protein level but up-regulated Bax protein level, suggesting that regulating mitochondria signaling pathway might be one potential mechanism of action of LMW-AP-FBG against CT-26 cells. 

5-FU, a thymidylate synthase inhibitor, can induce cytotoxicity by interfering with the synthesis of DNA and RNA [25,26]. It is an essential drug in the treatment of CRC due to a high response rate (40–50%) when it is combined with oxaliplatin, irinotecan, and other anticancer drugs [27]. The present study confirmed that 5-FU treatment significantly inhibited tumor growth in vivo by inhibiting proliferation and inducing apoptosis, although weight loss was observed. Unlike 5-FU, LMW-AP-FBG treatment did not reduce PCNA-positive cells in tumor tissues of CT-26 bearing mice, suggesting that LMW-AP-FBG treatment could not inhibit cell proliferation under an in vivo condition. However, when the viability of CT-26 cells was detected using CCK-8 in vitro, a significant decrease in viable cells was observed after LMW-AP-FBG treatment. Although reasons for these differences between in vivo and in vitro tests are currently unclear, apoptosis might be a major cause of decreased viability of CT-26 cells in vitro condition.

The NF-κB signaling pathway is a key regulator of inflammation. It is tightly associated with carcinogenesis. NF-κB has been shown to play a key role in multiple stages of malignancy development in the colon, ranging from formation of polyps to development of an invasive adenocarcinoma [28,29,30]. Abnormal activation of NF-κB has been found in approximately 50% of CRC patients [31]. It can promote the development of colitis-associated cancers [31]. A proinflammatory tumor microenvironment established in CRC can affect cell survival, proliferation, metastasis, and angiogenesis as a result of enhanced activation of NF-κB [31,32,33]. NF-κB can up-regulate the expression of anti-apoptotic proteins (such as Bcl-2 and Bcl-xL) and pro-inflammatory cytokines (such as TNFα, IL-6, and IL-1β) in CRC [31,34]. Previously, our study showed that LMW-AP-FBG treatment significantly blocked IκBα degradation and NF-κB phosphorylation in LPS-stimulated RAW264.7 cells, suggesting that LMW-AP-FBG could inhibit NF-κB activation [6]. Based on these results, we speculated that down-regulated NF-κB activation by LMW-AP-FBG might be one of the causes of the anticancer mechanism of LMW-AP-FBG.

The best-known receptor for β-1,3-1,6-glucan is Dectin-1, which is primarily expressed on a variety of immune cells [1,2,3]. Dectin-1 is widely expressed in the myeloid lineage, which involves macrophages/monocytes, dendritic cells and neutrophils, as well as in γδ T cells from the lymphoid lineage [35,36,37]. These broad expression patterns throughout the immune system suggest a highly complex variety of responses and signaling following Dectin-1 activation. After binding to its ligand, Dectin-1 causes the phosphorylation of immunoreceptor tyrosine-based activation-like motifs in its tyrosine residues [36,37]. Spleen associated tyrosine kinase (Syk) is then recruited to the two phosphorylated receptors, leading to the formation of a complex involving CARD9, BCL-10 and MALT1 [36,37]. This activated complex controls NF-κB activation. Another Syk downstream signal leads to activation of phospholipase Cγ2, which in turn activates MAPK-dependent and calcineurin-dependent pathways [37]. 

Several reports showed that binding of a β-glucan with a high molecular weight causes clustering of Dictin-1, which promotes inflammatory responses. On the other hand, small β-glucans also bind to Dictin-1, but do not induce clustering of the receptor [1,38]. When Dectin-1 acts as a receptor for LMW-AP-FBG in RAW264.7 cells, it activates the signaling pathways of NF-kB and MAPK, but previous studies using β-glucan with low molecular weight including ours have shown conflicting results [6,38]. These results suggest that LMW-AP-FBG may have distinct receptors and/or signaling pathways than Dectin-1 and Syk. 

As mention above, the main receptors of β-glucan, such as Dectin-1 and complement receptor 3, are mainly expressed in immune cells [35,36,37], and it is not clear whether these receptors are expressed in colon cancer cells. To date, there has been no report that Dectin-1 are expressed in CT-26 cells, which are mouse colorectal cancer cells used in this experiment. Therefore, more research is needed to fully understand the expression and regulation of β-glucan receptors in colorectal cancer cells. However, it can be inferred that some of the antitumor activity of LMW-AP-FBG in the mouse tumor model transplanted with CT-26 cells used in this study was achieved through the immunomodulation through these receptors.

Taken together, the present study demonstrates that LMW-AP-FBG can efficiently inhibit the viability of colon cancer cell by inducing apoptosis pathways via activation of caspases and disruption of mitochondrial membrane potential. Additionally, LMW-AP-FBG exhibited growth inhibitory effects on CT-26 cells in vivo (Figure 5). These findings suggest that LMW-AP-FBG could be potentially useful as an effective therapeutic agent or adjuvant for the treatment of CRC, although further studies are needed.

## Figures and Tables

**Figure 1 biomedicines-11-00529-f001:**
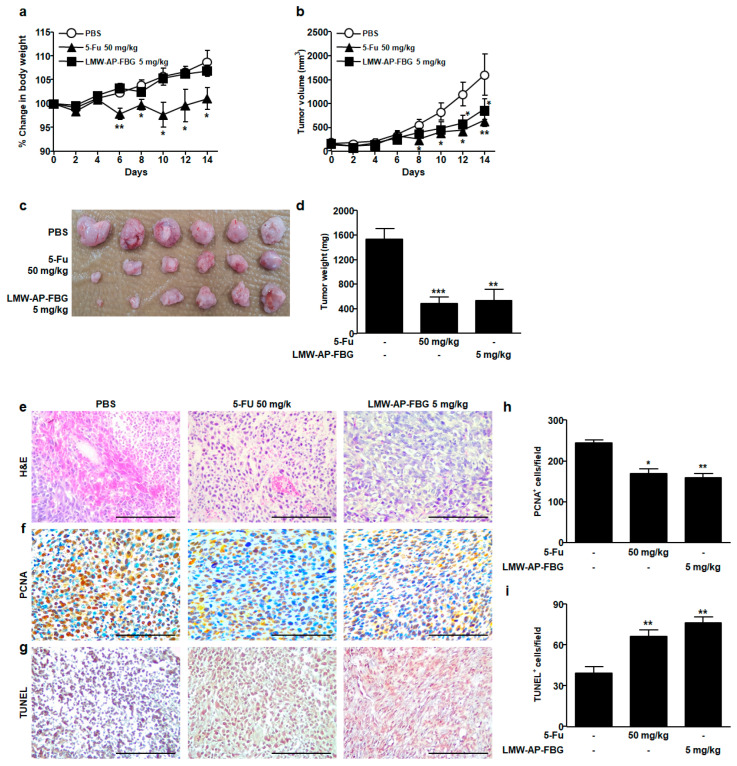
LMW-AP-FBG inhibits colon cancer progression in CT-26 tumor-bearing BALB/c mice. CT-26 murine colon cancer cells (1×10^6^ cells)/mouse were subcutaneously injected into BALB/c mice. When the tumor size reached approximately 100 mm^3^, mice were split into three groups (six BALB/c mice each). PBS and LMW-AP-FBG were administered intraperitoneally for 14 consecutive days. 5-FU was administered twice per weeks to the tumor-bearing mice. (**a**) Body weight and (**b**) tumor volume were measured every two days. After mice were sacrificed, solid tumors were separated. (**c**) Dissected tumors coming from mice were weighed and photographed and (**d**) weighed. (**e**) Paraffin sections of CT-26 tumor tissues were analyzed after H&E staining. (**f**,**h**) Expression and quantification of PCNA-positive staining in CT-26 tumor tissues were examined by IHC using ImageJ in three random fields. (**g**,**i**) TUNEL staining and quantification of TUNEL-positive cells in CT26 tumor tissues. Scale bar, 100 μm. Data are presented as mean ± SEM of six mice per group. * *p* < 0.05, *** p* < 0.01 *** *p* < 0.001 compared with PBS-treated control group.

**Figure 2 biomedicines-11-00529-f002:**
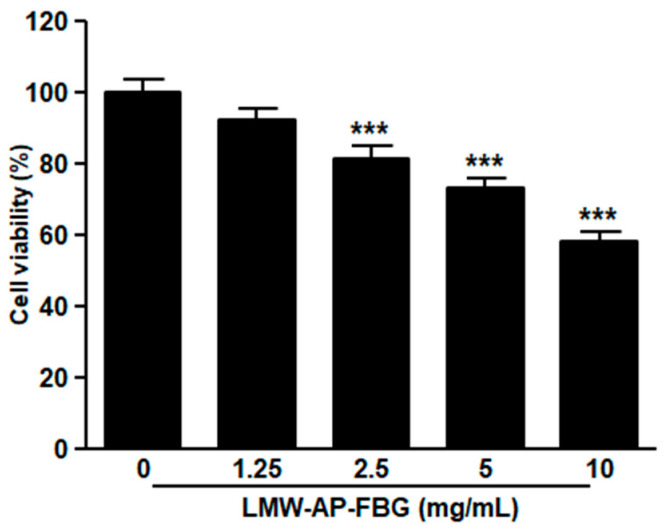
LMW-AP-FBG inhibits CT-26 cells viability. CT-26 cells were treated with indicated concentrations of LMW-AP-FBG for 24 h. Cell viability was determined by CCK-8 assay. Results are presented as mean ± SEM of three independent experiments for triplicate. *** *p* < 0.001 compared with untreated control cells (0 mg/mL).

**Figure 3 biomedicines-11-00529-f003:**
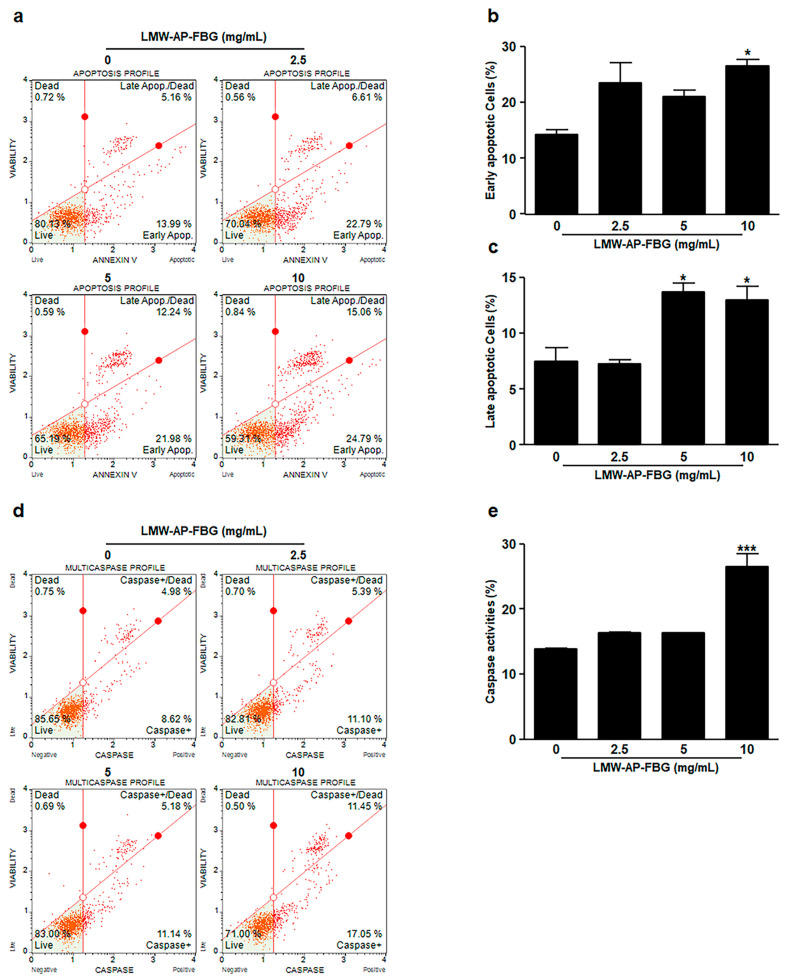
LMW-AP-FBG induces apoptosis in CT-26 cells. CT-26 cells were treated with indicated concentrations of LMW-AP-FBG for 24 h. Apoptosis was measured after staining cells with Annexin V and 7-AAD. (**a**) Representative annexin V dot plots are shown. (**b**) Percentages of cells in early and (**c**) late apoptosis in CT-26 cells are marked on a graph. (**d**) Representative dot plots of caspase activity measurement with a Muse™ Multi-caspase Kit. (**e**) Percentages of caspase activities were marked on a graph. Results are presented as mean ± SEM of three independent experiments for triplicate. * *p* < 0.05, *** *p* < 0.001, compared with untreated control cells (0 mg/mL).

**Figure 4 biomedicines-11-00529-f004:**
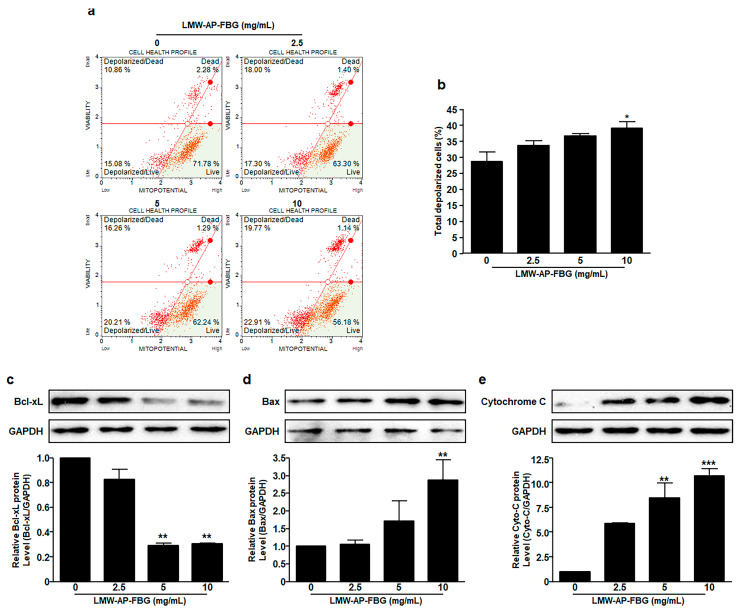
LMW-AP-FBG induces depolarization of mitochondrial membrane. CT-26 cells were treated with indicated concentrations of LMW-AP-FBG for 6 h. Mitochondrial membrane potential was assessed with a Muse Cell Analyzer using a Muse MitoPotential kit. (**a**) Typical profile plots are presented. (**b**) Percentage of total depolarized CT-26 cells treated with increasing doses of LMW-AP-FBG for 6 h. After cell lysis, equal amounts of proteins were separated by SDS-PAGE, transferred to PVDF membrane, and immunoblotted with antibodies against (**c**) Bcl-xL, (**d**) Bax, (**e**) cytochrome c, and GAPDH. Densitometric analysis was performed using ImageJ ver. 1.150i. Results are presented as mean ± SEM of three independent experiments. * *p* < 0.05, ** *p* < 0.01, *** *p* < 0.001 compared with untreated control cells (0 mg/mL).

**Figure 5 biomedicines-11-00529-f005:**
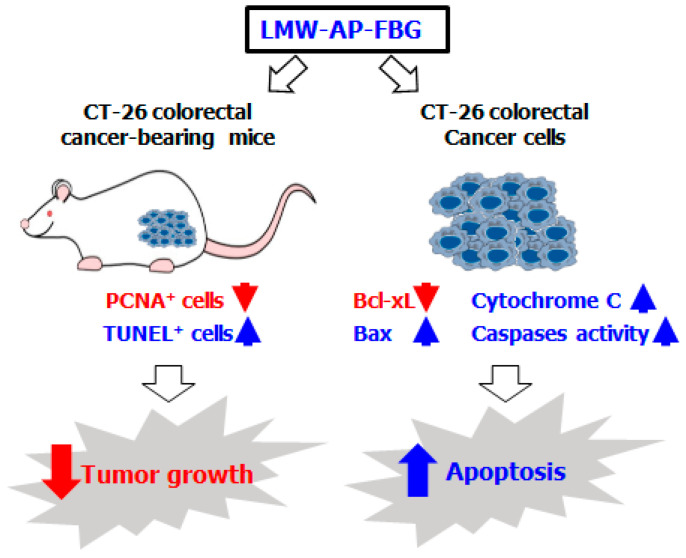
Schematic illustration of possible anticancer mechanisms responsible for the effects of LMW-AP-FBG against CT-26 colorectal cancer cells. CT-26 bearing mice treated with LMW-AP-FBG showed a significant decrease in PCNA-positive cells and an increase in TUNEL-positive cells in tumor tissue. CT-26 cells treated with LMW-AP-FBG showed anticancer activity by promoting apoptosis through increasing the ratio of Bax/Bcl-xl as well as activated caspase and cytosolic cytochrome c.

## Data Availability

The data presented in this study are available on request from the corresponding author.

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
