# Peer review of "Low-Molecular-Weight β-1,3-1,6-Glucan Derived from Aureobasidium pullulans Exhibits Anticancer Activity by Inducing Apoptosis in Colorectal Cancer Cells"

_biomedicines, 2023, doi:10.3390/biomedicines11020529_

Round 1
Reviewer 1 Report
Kim et al, propose β-glucan, a plant polysaccharide, as a new therapeutic drug against colorectal cancer. The authors observed that treatment with this agent shows anticancer properties both in vivo and in vitro. All in all, the manuscript is well written, executed and refined, with only minor comments to improve it.
·Although the study is interesting and well-constructed, it would be remarkable to investigate the effects that this substance has on a normal counterpart of the specific tumor here analyzed. Has a similar study ever been conducted to ascertain the specificity of the substance towards tumor tissues? and would it be possible to add such experiments to the manuscript?
·It would also be interesting to expand the introduction by adding a section dedicated to other natural substances with similar effects and characteristics. There are many other works that investigate substances of natural origin, for example: doi:10.3390/cancers11101456
· It would be better to insert a separate and final section, much more in-depth, on the conclusions drawn by the authors;
· The figures presented are well rendered and the data clearly visible; I have no further comments on this;
·The references listed are appropriate, it would be correct to add both new and suggested references for the introduction when this is expanded as required.
Author Response
Reviewer 1:
Kim et al, propose β-glucan, a plant polysaccharide, as a new therapeutic drug against colorectal cancer. The authors observed that treatment with this agent shows anticancer properties both in vivo and in vitro. All in all, the manuscript is well written, executed and refined, with only minor comments to improve it.
Although the study is interesting and well-constructed, it would be remarkable to investigate the effects that this substance has on a normal counterpart of the specific tumor here analyzed. Has a similar study ever been conducted to ascertain the specificity of the substance towards tumor tissues? and would it be possible to add such experiments to the manuscript?
Above all, I would like to thank you for your comments to improve the quality of this manuscript.
As you may notice, low-molecular-weight β-1,3-1,6-glucan extracted from Aureobasidium pullulans (LMW-AP-FBG) is a novel substance, and studies measuring its physiological and biological activity are extremely rare. Studies on β-1,3-1,6-glucan derived from A. pullulans (AP-FBG) are mostly for immune regulation [1-3], and research on antitumor activity is limited. Kimura et al. showed the antitumor and antimetastatic activities of AP-FBG using a mouse model in which CT26 cells were transplanted into the spleen [4]. Further studies showed that AP-FBG induced the expression of TNF-related apoptosis-inducing ligand in murine and human macrophage-like cells, indicating that AP-FBG-induced TRAIL expression stimulates tumor cell apoptosis [5]. Following the constructive advice of the reviewer, this content was inserted in the introduction of the revised manuscript.
In addition, as additional research to this study, experiments using human colon and ovarian cancer cells and in vivo xenograft mouse models are currently in progress, and the results of these studies will be announced soon.
[1] Tada R., Tanioka A., Iwasawa H., Hatashima K., Shoji Y., Ishibashi K.I., Adachi Y., Yamazaki M., Tsubaki K., Ohno N. Structural Characterisation and Biological Activities of a Unique Type Beta-D-Glucan Obtained From Glycoconj. J. 2008;25:851–861. doi: 10.1007/s10719-008-9147-3.
[3] Aureobasidium pullulansKimura Y., Sumiyoshi M., Suzuki T., Sakanaka M. Antitumor and Antimetastatic Activity of a Novel Water-Soluble Low Molecular Weight β-1,3-D-glucan (branch β-1,6) isolated from AP 1A1 strain black yeast. Anticancer Res. 2006;26:4131–4142.
[5] Kawata K., Iwai A., Muramatsu D., Aoki S., Uchiyama H., Okabe M., Hayakawa S., Takaoka A., Miyazaki T. Stimulation of Macrophages With the β-Glucan Produced by Aureobasidium pullulans Promotes the Secretion of Tumor Necrosis Factor-Related Apoptosis Inducing Ligand (TRAIL) PLoS ONE. 2015;10:e0124809. doi: 10.1371/journal.pone.0124809.
It would also be interesting to expand the introduction by adding a section dedicated to other natural substances with similar effects and characteristics. There are many other works that investigate substances of natural origin, for example: doi:10.3390/cancers11101456
Following the constructive advice of the reviewer, we added the antitumor effects of various polysaccharides with glucans at the introduction of the revised manuscript.
It would be better to insert a separate and final section, much more in-depth, on the conclusions drawn by the authors;
On the advice of the reviewer, we have inserted a separate final section in the concluding part of the revised manuscript.
The references listed are appropriate, it would be correct to add both new and suggested references for the introduction when this is expanded as required.
Newly added references were prepared and added according to the format.
Reviewer 2 Report
When presenting each individual result, indicate how many experiments, or animals, or wells are included in the analysis. In the statistical analysis paragraph authors have stated: mean ± SEM of results from at least three experiments.... Three experiments can be three times one animal, or one well with cells. It needs to be specified for each experiment.
P3. MMP is a common abbreviation used for Matrix metalloproteinase. Since this abbreviation is used only once after being introduced, I suggest avoiding the use of this abbreviation.
The authors should not draw a conclusion based on one trial that LMW-AP-FBG is a drug that can be used in the treatment of colorectal cancer.
Their conclusion can only be within their results: under the stated research conditions in cell culture and in a specific mouse model, LMW-AP-FBG showed an antitumor effect. Other animal studies, and in vitro studies on human cancer cell lines could confirm this effect, and LMW-AP-FBG could be tested in humans as an antitumor drug.
Author Response
Reviewer 2:
When presenting each individual result, indicate how many experiments, or animals, or wells are included in the analysis. In the statistical analysis paragraph authors have stated: mean ± SEM of results from at least three experiments.... Three experiments can be three times one animal, or one well with cells. It needs to be specified for each experiment.
We appreciate your comments to improve the quality of this manuscript. We corrected it in the revised manuscript.
P3. MMP is a common abbreviation used for Matrix metalloproteinase. Since this abbreviation is used only once after being introduced, I suggest avoiding the use of this abbreviation.
Following the reviewer's constructive advice, we did not use MMP as an abbreviation in revised manuscript.
The authors should not draw a conclusion based on one trial that LMW-AP-FBG is a drug that can be used in the treatment of colorectal cancer. Their conclusion can only be within their results: under the stated research conditions in cell culture and in a specific mouse model, LMW-AP-FBG showed an antitumor effect. Other animal studies, and in vitro studies on human cancer cell lines could confirm this effect, and LMW-AP-FBG could be tested in humans as an antitumor drug.
Thanks for pointing out an important issue. As you said, we are currently verifying antitumor activity in human colon and ovarian cancer cell lines. If meaningful results are obtained from these in vitro experiments, animal experiments will be conducted using xenograft model and the results will be announced. Therefore, the abstract and conclusion were changed in revised manuscript following the reviewer's constructive advice.
Reviewer 3 Report
In this research paper, the authors have studied the effect of Low-Molecular weight b—1,3-1,6- glucan (LMW-AP-FBG) derived Aureo basidium pullulans on anti-cancer activity in colorectal cancer cells. They showed that daily intraperitoneal injections of LMW-AP-FBG significantly tumor growth in mice bearing CT-26 tumors by tumor proliferation and apoptosis. LMW-AP-FBG treatment reduced the viability of CT-26 cells in a dose-dependent manner, and this was associated with loss of mitochondrial transmembrane potential and increased activated caspases. They showed that LMW-AP-FBG exhibits anti-cancer properties both in vitro and in vivo studies, and suggest that it can be used as a an anti-cancer agent against colorectal cancers.
Minor comments:
1. The authors could improve the manuscript by performing studies on how LMW-AP-FBG affects MAP kinase and NF-kB pathways to associate the findings with inhibition of tumor cell growth.
2. A model figure embedded in discussion will enhance the quality of manuscript in the discussion section.
3. Is any information available as to how LMW-AP-FBG affects cell signaling? Does it bind to any specific receptors? Is it the LMW-AP-FBG or its degraded derivative (metabolite)? Clarification is required.
Author Response
Reviewer 3:
The authors could improve the manuscript by performing studies on how LMW-AP-FBG affects MAP kinase and NF-kB pathways to associate the findings with inhibition of tumor cell growth.
Is any information available as to how LMW-AP-FBG affects cell signaling? Does it bind to any specific receptors? Is it the LMW-AP-FBG or its degraded derivative (metabolite)? Clarification is required.
We are grateful for timely comments that improve the quality of this manuscript and these two questions from the reviewer are related, so we'll answer them together.
As you may know, the best-known receptor for β-1,3-1,6-glucan is Dectin-1, which is primarily expressed on a variety of immune cells [1-3]. Dectin-1 is widely expressed in the myeloid lineage, which involves macrophages/monocytes, dendritic cells and neutrophils, as well as in γδ T cells from the lymphoid lineage [1-3]. These broad expression patterns throughout the immune system suggest a highly complex variety of responses and signaling following Dectin-1 activation. Most Dectin-1 trigger signaling pathways are well known in bone marrow cells such as monocytes, macrophages or DCs of mouse and human origin. After binding to its ligand, Dectin-1 causes the phosphorylation of immunoreceptor tyrosine-based activation (ITAM)-like motifs in its tyrosine residues. Spleen Associated Tyrosine Kinase (Syk) is then recruited to the two phosphorylated receptors, leading to the formation of a complex involving CARD9, BCL-10 and MALT1. This activated complex controls NF-κB activation. Another Syk downstream signal leads to activation of phospholipase Cγ2, which in turn activates mitogen-activated protein kinases (MAPKs)-dependent and calcineurin-dependent pathways.
Several reports showed that binding of a β-glucan with a high molecular weight causes clustering of Dictin-1, which promotes inflammatory responses. On the other hand, small β-glucans also bind to Dictin-1, but do not induce clustering of the receptor [4,5].
When Dectin-1 acts as a receptor for LMW-AP-FBG in RAW264.7 cells, it activates the signaling pathways of NF-kB and MAPK, but previous studies using β-glucan with low molecular weight including ours have shown conflicting results [5,6]. These results suggest that LMW-AP-FBG may have distinct receptors and/or signaling pathways than Dectin-1 and Syk.
As mention above, the main receptors of β-glucan, such as Dectin-1 and complement receptor 3, are mainly expressed in immune cells, and it is not clear whether these receptors are expressed in colon cancer cells. To date, there has been no report that Dectin-1 are expressed in CT-26 cells, which are mouse colorectal cancer cells used in this experiment. Therefore, more research is needed to fully understand the expression and regulation of β-glucan receptors in colorectal cancer cells. However, it can be inferred that some of the antitumor activity of β-glucan in the mouse tumor model transplanted with CT-26 cells used in this study was achieved through the regulation of immune activity through these receptors.
Therefore, the Discussion was modified by citing this study and other studies in the revised manuscript.
[1] Tsoni SV, Brown GD. beta-Glucans and dectin-1. Ann N Y Acad Sci. 2008 Nov;1143:45-60. doi: 10.1196/annals.1443.019.
[3] Suzuki T, Kusano K, Kondo N, Nishikawa K, Kuge T, Ohno N. Biological Activity of High-Purity β-1,3-1,6-Glucan Derived from the Black Yeast Aureobasidium pullulans: A Literature Review. Nutrients. 2021 Jan 16;13(1):242. doi: 10.3390/nu13010242.
[5] No H, Kim J, Seo CR, Lee DE, Kim JH, Kuge T, Mori T, Kimoto H, Kim JK. Anti-inflammatory effects of β-1,3-1,6-glucan derived from black yeast in RAW264.7 cells. Int J Biol Macromol. 2021 Dec 15;193(Pt A):592-600. doi: 10.1016/j.ijbiomac.2021.10.065.
A model figure embedded in discussion will enhance the quality of manuscript in the discussion section.
Following the reviewer's constructive advice, we added a figure of the model under discussion to the revised manuscript.
